# Glycemic control during TB treatment among Filipinos: The Starting Anti-Tuberculosis Treatment Cohort Study

**Lauren Oliveira Hashiguchi**[1,2], **Julius Patrick Ferrer**[3], **Shuichi Suzuki**[2], **Benjamin N. Faguer**[2], **Juan Antonio Solon**[3], **Mary Christine Castro**[3], **Koya Ariyoshi**[2], **Sharon E. Cox**[2,4,5☉], **Tansy Edwards**[2,6☉] *

1 Faculty of Public Health and Policy, London School of Hygiene & Tropical Medicine, London, United Kingdom, 2 School of Tropical Medicine & Global Health, Nagasaki University, Nagasaki, Japan, 3 Nutrition Center of the Philippines, Muntinlupa City, Manila, Philippines, 4 Faculty of Epidemiology & Population Health, London School of Hygiene & Tropical Medicine, London, United Kingdom, 5 Tuberculosis Unit, United Kingdom Health Security Agency, London, United Kingdom, 6 Medical Research Council International Statistics and Epidemiology Group, London School of Hygiene & Tropical Medicine, London, United Kingdom

☉ These authors contributed equally to this work.
* Tansy.Edwards@lshtm.ac.uk

**Data Availability Statement:** Analyses utilized data from the Starting Anti-TB Treatment (St-ATT) cohort (ISRCTN16347615). A minimized

## Abstract

Poor TB treatment outcomes are observed in patients with type 2 diabetes mellitus (DM) comorbidity and glycemic control throughout treatment may play a role. The objective of this study was to investigate glycemic control longitudinally among Filipino adults undergoing TB treatment using mixed-effects linear and logistic regression. Analyses were conducted in 188 DM-TB patients out of 901 enrolled in the Starting Anti-TB Treatment (St-ATT) cohort, with a median baseline glycosylated hemoglobin (HbA1c) of 8.2% (range 4.5–13.3%). Previous versus new DM diagnosis was associated with higher mean HbA1c (worse glycemic control) during treatment, with a smaller effect amongst those with central obesity (coefficient 0.80, 95% confidence interval [CI] 0.26, 1.57, $P$ = 0.043) than amongst those without central obesity (coefficient 3.48, 95% CI 2.16, 4.80, $P$<0.001). In those with a new DM diagnosis, central obesity was associated with higher blood glucose (coefficient 1.62, 95% CI 0.72, 2.53, P = 0.009). Of 177 participants with ≥2 HbA1c results, 40% had uncontrolled glycemia (≥2 HbA1c results ≥8%). Of 165 participants with ≥3 HbA1c results, 29.9% had consistently-controlled glycemia, 15.3% had initially-uncontrolled glycemia, and 18.6% had consistently-uncontrolled glycemia. Previous versus new DM diagnosis and glucose-lowering medication use versus no use were associated with having uncontrolled versus controlled glycemia (adjusted odds ratio [aOR] 2.50 95%CI 1.61, 6.05, P = 0.042; aOR 4.78 95% CI 1.61,14.23, P<0.001) and more likely to have consistently-uncontrolled versus consistently-controlled glycemia (adjusted relative risk ratio [aRRR] 5.14 95% CI 1.37, 19.20, P = 0.015; aRRR 10.24 95% CI 0.07, 0.95, P = 0.003). Relapse cases of TB were less likely than new cases to have uncontrolled (aOR 0.20 95%CI 0.06, 0.63, P = 0.031) or consistently-uncontrolled (aRRR 0.25 95%CI 0.07, 0.95, P = 0.042) versus controlled glycemia. Those with long-term DM, suggested by previous diagnosis, glucose-lowering medication

anonymized dataset will be made freely available as a dataset linked to the project found at the DOI link, deposited in LSHTM data compass with the following DOI: https://doi.org/10.17037/DATA. 00002313.

**Funding:** This work was supported by the Nagasaki University "Doctoral Program for World-leading Innovative and Smart Education" for Global Health, also referred to as the WISE Programme (no grant number, awarded to LOH), and a research grant, "Kakenhi" (17H04662 to SEC), from the Ministry of Education, Culture, Sports, Science & Technology, Japan. TE receives salary funding from a joint grant award funded by the UK Medical Research Council (MRC) and the UK Department for International Development (DFID) under the MRC/DFID Concordat agreement and is also part of the EDCTP2 programme supported by the European Union: Grant Ref: MR/R010161/1. The funders had no role in study design, data collection and analysis, decision to publish, or preparation of the manuscript.

**Competing interests:** The authors have declared that no competing interests exist.

use and possibly central obesity, may require additional support to manage blood glucose during TB treatment.

## Introduction

People with diabetes mellitus (DM), a chronic, metabolic disorder characterized by hyperglycemia, are at greater risk of having active tuberculosis (TB) disease and experiencing poor TB treatment outcomes compared to those without DM [1–8]. Altered immune function plays a role in the increased risk of developing active TB disease observed among diabetic persons, but the exact mechanisms are not yet understood [9–11]. A 2019 systematic review of 64 studies estimated that TB patients with DM comorbidity (TB-DM) have nearly twice the odds of death compared to TB patients without DM [8].

As the global burden of DM grows, managing dual TB-DM disease is increasingly important. In the Western Pacific, DM prevalence may increase to 260 million people by 2045 [12,13]. Many countries in this region have a high TB burden, including the Philippines where TB incidence is one of the highest globally (539/100,000) [14]. In 2020, the International Diabetes Federation estimated DM affects 6.5% of Filipino adults [13].

Though studies have established that DM is associated with poor TB treatment outcomes [1–5], less is known about the role of DM management, which can be monitored by glycemic control. Long-term poor glycemic control, assessed through glycosylated hemoglobin (HbA1c) seems to have a role in increased risk of TB, severity of clinical presentation of TB, and poor TB treatment outcomes [15–18]. Glycemic control among persons with DM is influenced by factors related to the individual and their disease, their management practices, and the health system they receive care within [19,20]. In particular, central obesity—fat deposition in the abdomen—is prevalent in Asian countries and a predictor of diabetes [21] which is associated with hyperglycemia [22–26].

Previous studies have found that poor glycemic control—assessed at the beginning of TB treatment—is associated with poor TB treatment outcomes [27–29]. TB-associated inflammation and TB-specific insulin resistance can provoke transient hyperglycemia—a temporary increase in blood glucose which resolves with effective TB treatment [30–34], which may confound previous findings. Degree of glycemic control may be more important in determining TB treatment outcomes than the absence or presence of diabetes [18,27,28,30,35,36]. Characterizing patterns of glycemic control and their predictors is of interest in understanding the relationship between DM and adverse TB treatment outcomes.

To our knowledge, there is only one other analysis of longitudinal HbA1c data which explores associations with glycemic control during TB treatment [37] and only one study of glycemic control among TB-DM populations specifically excludes those with suspected transient hyperglycemia [30]. There are little published data about TB-DM from Southeast Asia, especially from the Philippines. A 2017 systematic review by Zheng, et al. [38] of diabetes and TB in Asian countries identified eight epidemiological studies exploring the prevalence of DM among TB populations [39–43]. Few studies have examined glycemic control among TB-DM patients in Southeast Asia [44].

The aim of this study was to investigate glycemic control among DM patients receiving Directly Observed Treatment for the Treatment of TB (TB-DOTS) in the Philippines and the factors influencing glycemic control, in order to generate evidence to inform and improve targeted support for DM management in the context of limited resources.

## Material and methods

### Study design and setting

This study was part of the Starting Anti-TB Treatment (St-ATT) cohort, a prospective cohort study of people starting treatment for drug-sensitive and drug-resistant TB in public TB-DOTS clinics in the Philippines (ISRCTN16347615) [45]. Participants were recruited from 13 clinics in Metro Manila (N = 3), Cebu (N = 5) and Negros Occidental (N = 5) between August 2018 and February 2020. Four sites were centers for the programmatic management of drug resistant TB (MDR-TB). Clinics in Cebu and Negros Occidental were in urban, peri-urban and rural areas, and clinics in Manila were in urban areas.

### Participants

Non-pregnant adults (≥18 years) who were enrolled at participating sites, with either a previous diagnosis of DM or a new diagnosis (Table 1), and with at least one HbA1c measurement collected. Participants with transient hyperglycemia (Table 1) were excluded from the analysis. Informed consent procedures were performed by St-ATT research nurses. Written informed consent was obtained from all participants prior to participation in the local language (Filipino, Cebuano, Hiligaynon) or English. St-ATT research nurses completed informed consent procedures—reading consent forms verbally—in quiet and private locations to ensure patients had free and voluntary consent uninfluenced by health center staff.

### Ethical approval

All methods for this research involving human subjects were carried out in accordance with the relevant guidelines and regulations laid out in the Declaration of Helsinki. This study protocol was approved by the Philippines-accredited ethical review board, the Asian Eye Institute of the Philippines in Manila (REF 2020–00), by the London School of Hygiene and Tropical

**Table 1. Starting Anti-Tuberculosis Treatment (St-ATT) Cohort Study diagnostic criteria for diabetes mellitus (DM).**

| Criteria | Definition | |
|---|---|---|
| **Glycosylated hemoglobin (HbA1c) cut-points** | ≥6.5% | The cut-point for diagnosing DM in presence of DM symptoms recommended by an international expert committee [46]. |
| | >8% | The cut-point for poor blood glucose control for tuberculosis (TB) patients with a DM comorbidity (TB-DM) recommended by the International Union Against TB and Lung Disease [15]. |
| | >10% | Severe hyperglycemia among persons with TB-DM recommended by the International Union Against TB and Lung Disease [15]. |
| **Previously diagnosed DM** | Self-reported previous DM diagnosis by a physician AND if no HbA1c values ≥6.5%, self-reported use of standard glucose-lowering medication (insulin, gliclazide, metformin) | |
| **Newly diagnosed, suspected DM** | No previous DM diagnosis at enrollment AND either: HbA1c test result ≥6.5% at enrollment or during treatment, OR self-reported physician DM diagnosis after enrollment. | |
| **Transient hyperglycemia** | No self-reported physician diagnosis of DM or glucose-lowering medication use during TB treatment AND: HbA1c result ≥6.5% at enrollment only, OR HbA1c <6.5% at baseline and no subsequent HbA1c test result exceeds 10%[a] | |
| **Confirmed DM** | Previous or new diagnosis as above, excluding transient hyperglycemia. | |

[a] In the absence of official guidance on how to determine transient hyperglycaemia among TB patients using HbA1c data, the definition applied in this analysis was developed by the researchers, drawing from global TB-DM treatment guidelines from the Global Union Against TB and Lung Disease [15].

Medicine in London, United Kingdom (17839), Nagasaki University, School of Tropical Medicine and Global Health in Nagasaki, Japan (NU_TMGH_2020_1010_2), and by two participating hospitals in the Philippines: Riverside Health Centre ethics board in Bacolod City, Negros (DPOTMH-REC 2020–05), and San Lazaro Hospital in Manila, National Capital Region (SLH-RERU-2020-012-E). During the COVID-19 pandemic, protocol was modified to allow phone interview, and this was approved by all ethics committees that approved the study.

## Data collection

Trained research nurses completed study assessments using direct electronic data capture into Open Data Kit software. Assessments were conducted using structured questionnaires administered through participant interview, fingerpick blood sampling for glycosylated hemoglobin (Point of Care, Trinity Biotech), and data extraction from participants' National TB Program treatment cards [45]. Data were uploaded to a secure server daily. Participants were followed at monthly appointments until treatment exit (typically 6–12 months for drug-sensitive TB [DS-TB]). Glycosylated hemoglobin was measured approximately tri-monthly after baseline. From February 2020 participants were followed by phone or home visit when in-person appointments were not possible due to COIVD-19 community restrictions. Data collection continued until the end of the study in October 2021.

## Outcomes

HbA1c (%) was analyzed as a repeated continuous outcome measure, and using a cut-point of 8% to determine hyperglycemia, as a binary outcome indicator of controlled vs uncontrolled glycemia and as a categorical outcome of degree of glycemic control during TB treatment (Table 2).

## Exposures

A literature review of the factors associated with glycemic control among TB patients with a diabetes comorbidity guided the inclusion of variables in the analysis.

Sociodemographic variables at start of TB treatment were: age (continuous), sex, region and residential area, marital status, membership to any of Philippines Health Insurance plans (Yes/No), employment (Yes/No), higher than primary school education (Yes/No), and absolute annual household income ($<$ 5,000 Philippine peso (PHP), 5000–9999 PHP, $>$10,000 PHP).

Research nurses conducted anthropometric measurements (enrollment and monthly) [47], including weight (to the nearest 0.1 kg; Seca 803 Clara Digital Personal Non-Medical Scale)

**Table 2. Description of study outcome measures.**

| Outcome measure | Description |
|---|---|
| **Continuous outcome: Glycosylated hemoglobin (HbA1c)** | Diabetes Control and Complications Trial; unit, expressed as a percentage. Measured at time of enrollment in the Starting Anti-TB Treatment Cohort Study and then approximately trimonthly. |
| **Binary outcome: Controlled and uncontrolled** | Amongst those with $\geq$2 HbA1c test results:<br>(i) Controlled: at least two HbA1c results $<$ 8%<br>(ii) Uncontrolled: $\geq$2 HbA1c results $\geq$ 8% |
| **Categorical outcome: Degree of glycemic control** | Amongst those with $\geq$3 HbA1c test results:<br>(i) Controlled: all HbA1c results $<$ 8%<br>(ii) Initially-uncontrolled: baseline HbA1c $\geq$ 8%, and all subsequent results $<$ 8%<br>(iii) Consistently-uncontrolled: all HbA1c results $\geq$ 8% |

and height (to the nearest 0.5 cm; Seca 216 Mechanical Stadiometer), measured on a flat surface with the patient upright, unassisted without shoes. Body mass index (BMI) was categorized for analysis according to World Health Organization (WHO) criteria for adults; underweight (BMI<18.5 kg/m$^{2)}$, normal (BMI 18.5–25.0), overweight (25.0–29.9), obese (BMI ≥30) [48]. Waist and hip circumference were measured at enrollment (to the nearest 0.5cm; Seca 201 measuring tape); waist-to-hip ratio (WHR) was calculated by dividing waist circumference by hip circumference. WHR was used to distinguish central obesity using WHO-recommended cut-point for diagnosing metabolic syndrome [49]: >0.85 for women and >0.9 for men. Blood pressure (BP) was measured at monthly appointments twice 5 minutes apart with the participant seated and at rest using an automated BP monitor (Omron HEM-907, Kyoto, Japan). When measurements were >5 mm Hg apart, a third measure was taken and the average of the two closest values was used. BP was categorized for analysis using the American College of Cardiology and American Heart Association guidelines [50]; normal = systolic BP (SBP) <120 mm Hg and diastolic BP (DBP) <80 mm Hg); elevated = SBP 120–129 and DBP<80); Stage 1 hypertension = SBP 130–139 and DBP 80–89; and Stage 2 hypertension = SBP ≥140 and DBP ≥90.

Variables capturing TB-related characteristics at enrollment included: TB treatment facility (public hospital, city health center, or rural health unit), method of TB diagnosis (clinically diagnosed, bacteriologically confirmed), new versus relapse TB case, duration of TB symptoms prior to treatment (weeks), TB treatment regimen (DS-TB or MDR-TB) and current TB symptoms (report of fever, cough, chills, weight loss, reduced appetite, hemoptysis, night sweats, or chest pain). A score of adherence to TB treatment in the past seven days was measured monthly using the eight-question Morisky Medication *Adherence* Scale (*MMAS*-8); high adherence = 8, medium = 6–7, and low <6 [51].

Self-reported current use of gliclazide, insulin, or metformin was captured at enrollment and every three months along with current DM complications assessed using a survey instrument based on that used by the Concurrent Tuberculosis and Diabetes Mellitus Consortium [52] and the Michigan Neuropathy Screening Instrument [53].

## Study size

The sample size for this study was determined by the primary objective of the St-ATT study, which was to investigate associations between DM and undernutrition with TB treatment outcomes [47]. The St-ATT cohort required a minimum of 800 participants for at least 90% power and 5% significance to detect associations between DM at the start of treatment with adverse TB treatment outcome, assuming 10–20% of the cohort would experience an adverse treatment outcome and 10–12% would have DM.

## Statistical methods

Characteristics of participants were first tabulated overall and by region. Differences in categorical characteristic variables by region were tested using Fisher's exact test, and differences between continuous variables were tested using an analysis of variance (ANOVA) test.

## Associations with HbA1c over time during TB treatment

Univariable associations with HbA1c at any point in time were investigated using mixed-effects linear regression with a random intercept for individuals and a random slope for time (days from the start of treatment). Visual inspection showed that HbA1c % decreased on average as treatment progressed, but the rate of decrease was non-linear. Therefore, time was transformed using a square root function with a constant of 0.01 added to achieve positivity of

values [54]. Differential changes in HbA1c over time within strata of categorical exposures were investigated by fitting an interaction between time and each exposure. Restricted maximum likelihood (REML) was the likelihood estimator, being more appropriate for smaller samples. Likelihood Ratio Test (LRT) of fixed effects cannot be used with models fit using REML [55]; a global Wald test with a small-sample adjustment for fixed effects [56] (P<0.1) was used to assess associations.

A multivariable model was developed using forward step-wise selection of variables in four blocks, starting with socio-demographic characteristics, followed by anthropometric variables, then DM-related variables, and TB-related variables. Central obesity was included in the final model a priori. Variables in each block were tested for inclusion in the order of strength of effect in univariable analysis. Selection of a final model was based on inclusion of factors associated with the outcome based on a Wald test P value of <0.1 in univariable analysis and retained if meeting this criterion after adjustment for other covariates. After adjustment for all factors remaining in the multivariable model, interactions terms between time and each covariate, and between covariates were tested.

### Analysis of predictors of poor glycemic control

Associations between participant characteristics and poor glycemic control were investigated using logistic and multinomial regression for the binary and categorical outcomes (Table 2), respectively. The same multivariable model-building approach described above was also used here. Logistic regression was used to investigate associations with controlled versus uncontrolled glycemia. Multinomial regression was used to investigate associations with the degree of poor glycemic control (i.e., controlled versus initially-uncontrolled and consistently-uncontrolled glycemia). Between-site variation (clustering) was investigated by comparing multivariable models with and without a random intercept for site. All data were analyzed with Stata (Version 15, College Station, Texas: StataCorp LP).

## Results

### Participants

Of 901 participants enrolled in the St-ATT cohort [47], 200 had previously diagnosed or newly diagnosed DM per St-ATT case definitions (Table 1, Fig 1). One person previously-diagnosed with DM with no HbA1c measurements was excluded. A further 48 suspected DM cases were identified as a result of external reported diagnoses or HbA1c results during follow-up. Fifty-nine suspected DM cases, mostly diagnosed at the time of enrollment, were transiently hyperglycemic (Fig 1) and excluded from analyses.

Among the 188 participants included in this study (Fig 1), there was an average of three HbA1c results per patient. More than half of the sample (N = 97) had data for baseline, three- and six-months (Table 3, S1 Fig). More participants in Manila had only one HbA1c measurement compared to other regions (Table 3, Fisher's test P<0.001).

The random-effects regression analysis sample included 188 participants with 482 HbA1c results. Nearly 70% (131/188) were male, the median age was 50 years, 78.2% (147/188) were receiving treatment for drug-sensitive TB (Table 3). Approximately half (99/188) were newly diagnosed with DM at the start of TB treatment. Compared to participants in Manila and Cebu, more participants in Negros Occidental had a lower income, health insurance, a new DM diagnosis, and longer duration of TB symptoms prior to starting TB treatment (P<0.05). Participants in Manila were more likely to be hypertensive or previously diagnosed with DM compared to other regions (P<0.05).

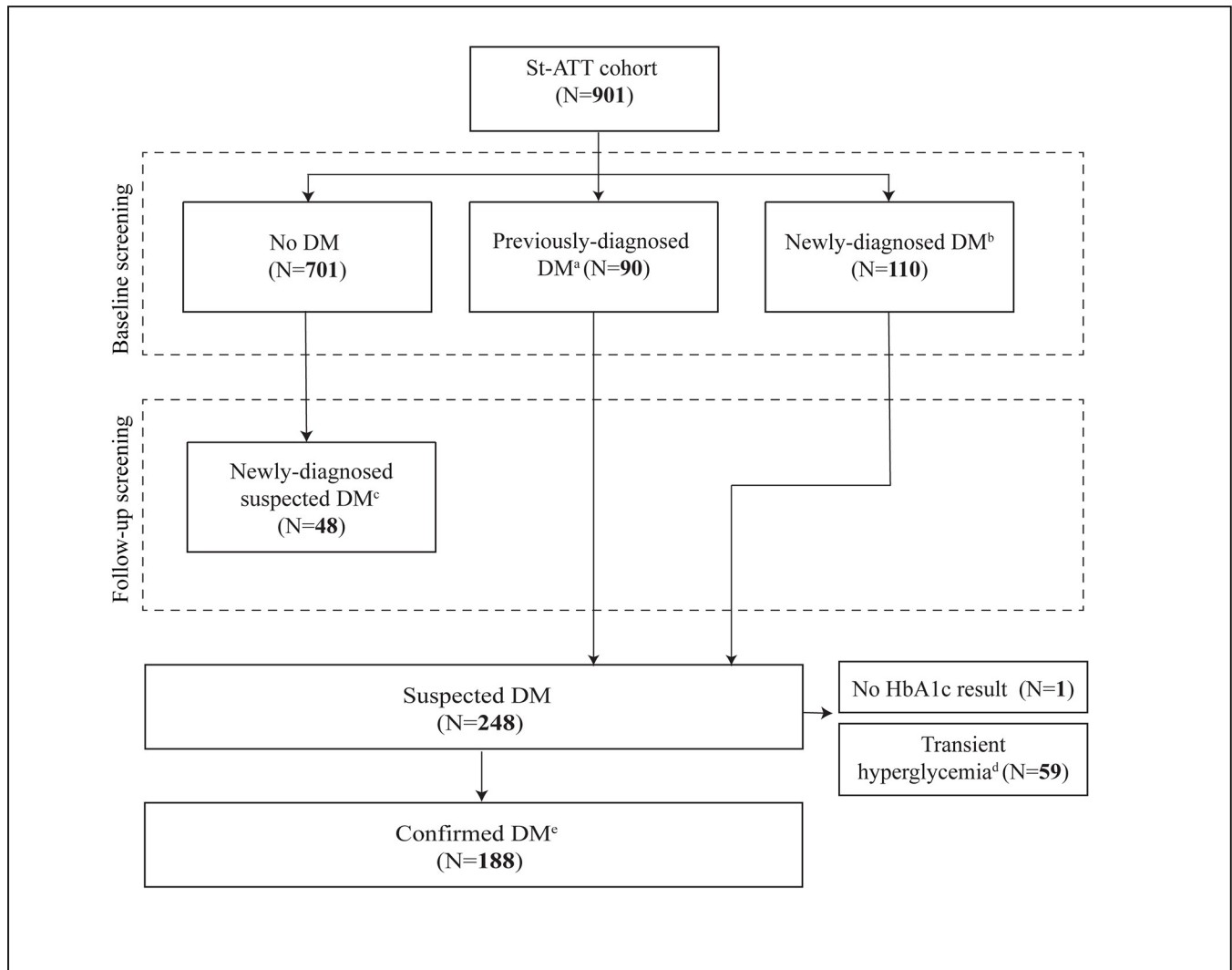

**Fig 1. Flowchart of diabetic persons within Anti-Tuberculosis Treatment Cohort and analysis population.** Abbreviations: Diabetes mellitus, DM; glycosylated hemoglobin, HbA1c, Starting Anti-TB Treatment Cohort Study.[a] Participant HbA1c measurement was ≥6.5% (the recognized cut-point for diagnosing DM in presence of DM symptoms) at the time of enrollment. [b] At enrollment, self-reported having been previously diagnosed with DM by a physician and, if HbA1c was missing or <6.5% at enrollment, to be using a standard glucose-lowering medication (insulin, gliclazide, metformin). [c] Participant had an HbA1c result ≥6.5% after enrollment. Of the 48 participants with an HbA1c ≥6.5%, 34 additionally reported having been diagnosed with DM by a physician (regardless of reported glucose-lowering medication use) since their last St-ATT appointment. [d] Participant never reported outside doctor diagnosis of DM or use of DM medication, and had either 1) a HbA1c ≥6.5% test result at baseline and no subsequent HbA1c values ≥ 6.5%; or 2) a HbA1c <6.5% test result at baseline and only one additional HbA1c test result which was <10% (threshold for severe hyperglycemia recommended by the International Union Against TB and Lung Disease [15]) during treatment. 48 of the 59 suspected DM cases who were transiently hyperglycemic were diagnosed at the time of enrollment into TB treatment. [e] The analyses of glycemic control include newly-diagnosed DM cases who did not have transient hyperglycemia in addition to previously diagnosed DM cases.

### Glycemic control

Of those with ≥2 HbA1c results, 85 persons had uncontrolled glycemia and 66 had controlled glycemia. Of those with ≥3 HbA1c results, 53 had consistently-controlled glycemia, 27 had initially-uncontrolled glycemia, and 33 had consistently-uncontrolled glycemia. Not everyone with a sufficient number of HbA1c results could be categorized into the glycemic control definitions (Table 2). More participants excluded from the binary and categorical outcome analysis populations were previously diagnosed with DM, lived in Manila and were hypertensive at

**Table 3. Socio-demographic, anthropometric, tuberculosis- and diabetes mellitus-related characteristics of 188 Starting Anti-TB Treatment Cohort Study participants with a diabetes mellitus (DM) comorbidity by region.**

| | Total, % (n = 188) | Negros Occidental, % (n = 77) | Cebu, % (n = 83) | Manila, % (n = 28) | P-value[a] |
|---|---|---|---|---|---|
| Age, years | | | | | 0.712 |
| Mean, SD | 50.6, 12.6 | 50.3, 13.5 | 50.4, 12.1 | 52.5, 11.8 | |
| Sex | | | | | 0.213 |
| Male | 131 (69.7) | 49 (63.6) | 63 (75.9) | 19 (67.9) | |
| Absolute annual household income level | | | | | < 0.001 |
| Less than 5,000 PHP | 73 (39.0) | 46 (59.7) | 19 (22.9) | 8 (29.6) | |
| 5000, 9999 PHP | 50 (26.7) | 20 (26.0) | 24 (28.9) | 6 (22.2) | |
| >10,000 PHP | 64 (34.2) | 11 (14.3) | 40 (48.2) | 13 (48.1) | |
| Marital status | | | | | 0.511 |
| Single | 37 (19.7) | 18 (23.4) | 13 (15.7) | 6 (21.4) | |
| Married | 134 (71.3) | 51 (66.2) | 64 (77.1) | 19 (67.9) | |
| Divorced/separated | 3 (1.6) | 2 (2.6) | 0 | 1 (3.6) | |
| Widowed | 14 (7.4) | 6 (7.8) | 6 (7.2) | 2 (7.1) | |
| Unemployed | 123 (65.4) | 54 (70.1) | 50 (60.2) | 19 (67.9) | 0.400 |
| Higher than primary school education | 44 (23.5) | 18 (23.4) | 21 (25.6) | 5 (17.9) | 0.733 |
| Possess health insurance[b] | 114 (66.3) | 52 (76.5) | 42 (53.8) | 20 (76.9) | 0.006 |
| Residential area | | | | | < 0.001 |
| Urban | 48 (25.5) | 11 (14.3) | 9 (10.8) | 28 (100.0) | |
| Peri-urban | 86 (45.7) | 34 (44.2) | 52 (62.7) | 0 | |
| Rural | 54 (28.7) | 32 (41.6) | 22 (26.5) | 0 | |
| Central obesity[c] | 131 (69.7) | 58 (75.3) | 52 (62.7) | 21 (75.0) | 0.184 |
| Blood pressure at baseline[d] | | | | | 0.067 |
| Normal | 73 (42.7) | 27 (44.3) | 39 (47.6) | 7 (25.0) | |
| Elevated | 16 (9.4) | 8 (13.1) | 7 (8.5) | 1 (3.6) | |
| Stage 1 Hypertension | 47 (27.5) | 17 (27.9) | 22 (26.8) | 8 (28.6) | |
| Stage 2 Hypertension | 35 (20.5) | 9 (14.8) | 14 (17.1) | 12 (42.9) | |
| BMI classification[e] | | | | | 0.155 |
| Underweight | 114 (61.0) | 49 (63.6) | 47 (57.3) | 18 (64.3) | |
| Normal | 45 (24.1) | 21 (27.3) | 21 (25.6) | 3 (10.7) | |
| Overweight | 26 (13.9) | 6 (7.8) | 13 (15.9) | 7 (25.0) | |
| Obese | 2 (1.1) | 1 (1.3) | 1 (1.2) | 0 | |
| Type of TB treatment facility | | | | | < 0.001 |
| Public Hospital | 51 (27.3) | 11 (14.3) | 20 (24.4) | 20 (71.4) | |
| City Health Center | 82 (43.9) | 34 (44.2) | 40 (48.8) | 8 (28.6) | |
| Rural Health Unit | 54 (28.9) | 32 (41.6) | 22 (26.8) | 0 (0.0) | |
| TB regimen | | | | | 0.478 |
| Drug sensitive | 147 (78.2) | 61 (79.2) | 62 (74.7) | 24 (85.7) | |
| Drug resistant | 41 (21.8) | 16 (20.8) | 21 (25.3) | 4 (14.3) | |
| New versus relapse TB case | | | | | 0.301 |
| New | 119 (63.6) | 45 (58.4) | 53 (64.6) | 21 (75.0) | |
| Basis of TB diagnosis | | | | | 0.577 |
| Clinically diagnosed | 62 (33.2) | 29 (37.7) | 25 (30.5) | 8 (28.6) | |
| Bacteriologically confirmed[f] | 125 (66.8) | 48 (62.3) | 57 (69.5) | 20 (71.4) | |
| Duration of TB symptoms prior to treatment (weeks) | | | | | 0.002 |
| Mean, SD | 8.4, 8.5 | 11.1, 11.3 | 6.6, 5.1 | 6.5, 4.9 | |

*(Continued)*

**Table 3.** (Continued)

| | Total, % (n = 188) | Negros Occidental, % (n = 77) | Cebu, % (n = 83) | Manila, % (n = 28) | P-value[a] |
|---|---|---|---|---|---|
| Median, Range | 6.0 0.0, 80.7 | 8.0 1.4, 80.7 | 5.0 0.0, 28.1 | 5.6 0.0, 25.9 | |
| Timing of DM diagnosis | | | | | 0.004 |
| Newly diagnosed | 99 (52.7) | 51 (66.2) | 39 (47.0) | 9 (32.1) | |
| Previously diagnosed | 89 (47.3) | 26 (33.8) | 44 (53.0) | 19 (67.9) | |
| Report of glucose-lowering medications during TB treatment[g] | 132 (70.6) | 47 (61.0%) | 60 (73.2%) | 25 (89.3%) | 0.017 |
| Report of any DM complication[h] during TB treatment | 44 (23.4) | 21 (27.3) | 16 (19.3) | 7 (25.0) | 0.490 |
| HbA1c at enrollment, % | | | | | |
| Mean, SD | 8.4 (2.2) | 8.0 (2.1) | 8.4 (2.6) | 9.5 (2.2) | 0.878 |
| Median, Range | 8.2 (4.5, 13.2) | 7.6 (5.1, 14.0) | 7.5 (4.5, 14.1) | 9.5 (5.8, 13.2) | |
| Number of HbA1c results | | | | | <0.001 |
| Baseline only | 22 (12.4) | 2 (2.8) | 11 (13.8) | 9 (34.6) | |
| Baseline & 3 months only | 25 (14.0) | 11 (15.3) | 11 (13.8) | 3 (11.5) | |
| Baseline, 3, & 6 months | 97 (54.5) | 46 (63.9) | 47 (58.8) | 4 (15.4) | |
| Baseline & 6 months only | 19 (10.7) | 7 (9.7) | 2 (2.5) | 10 (38.5) | |
| Baseline, 3, 6, 9 months | 12 (6.7) | 3 (4.2) | 9 (11.2) | 0 | |
| Baseline & 9 months only | 3 (1.7) | 3 (4.2) | 0 | 0 | |

BMI, Body Mass Index; CI, confidence interval; DBP, diastolic blood pressure; DM, diabetes mellitus; HbA1c, glycosylated hemoglobin; PHP, Philippine peso; St-ATT, Starting Anti-TB Treatment Cohort Study; SBP, systolic blood pressure; SD, standard deviation; TANDEM, Concurrent Tuberculosis and Diabetes Mellitus Consortium; TB, tuberculosis; WHO, World Health Organization.

[a] Fisher's exact test for categorical variables, ANOVA for continuous variables.

[b] Any of Philippines Health Insurance plan, Social Security, or Government Service Insurance.

[c] waist-to-hip ratio >0.85 for women, >0.9 for men as per WHO diagnostic criteria for metabolic syndrome [49].

[d] Normal (SBP <120 SBP mm Hg and DBP <80 mm Hg); elevated (SBP 120–129 and DBP <80); Stage 1 Hypertension (SBP 130–139 and DBP 80–89); and Stage 2 Hypertension (SBP ≥ 140 and DBP ≥ 90), by the 2017 American College of Cardiology and American Heart Association guidelines [50].

[e] BMI according to WHO criteria for adults: Underweight (BMI<18.5 kg/m$^2$), normal (BMI 18.5–25.0), overweight (25.0–29.9), obese (BMI ≥30) [48].

[f] Confirmed by GeneXpert (Cepheid), a cartridge-based nucleic acid amplification test for simultaneous rapid tuberculosis diagnosis and rapid antibiotic sensitivity test, or by direct sputum smear microscopy.

[g] Metformin, gliclazide, or insulin use self-reported at enrollment or any point during TB treatment.

[h] After enrollment in TB treatment, report of experiencing any of the TANDEM study DM complications [52]: Ever lost a limb or digit not through trauma, ever had a bypass or stenting surgery in limbs, non-healing wound for three or more months, heart attack, stroke, bypass or stenting heart surgery, diagnosis of angina or heart failure, cataract or laser eye surgery, glaucoma, acquired blindness not due to trauma, difficulty seeing or disturbed vision, renal failure. Additionally, the measure captures if participant had any symptom of distal symmetrical peripheral neuropathy using the Michigan Neuropathy Screening Instrument [53].

enrollment (P<0.05, S1 Table). Individual HbA1c trajectories are depicted by glycemic control pattern and timing of DM diagnosis (Figs 2 and 3).

## Associations with HbA1c during TB treatment

Region was significantly associated with HbA1c in univariable analysis (S3 Table) and was added into the model last to avoid adjusting for region rather than characteristics which explained variability in the data due to region. The final multivariable model (Table 4) adjusted for central obesity at baseline, blood pressure at baseline, timing of DM diagnosis, and interaction terms between central obesity and timing of DM diagnosis (P<0.001), and between time and TB treatment regimen (P = 0.096). There was no evidence of between-site variation (clustering, LRT P-value = 0.240).

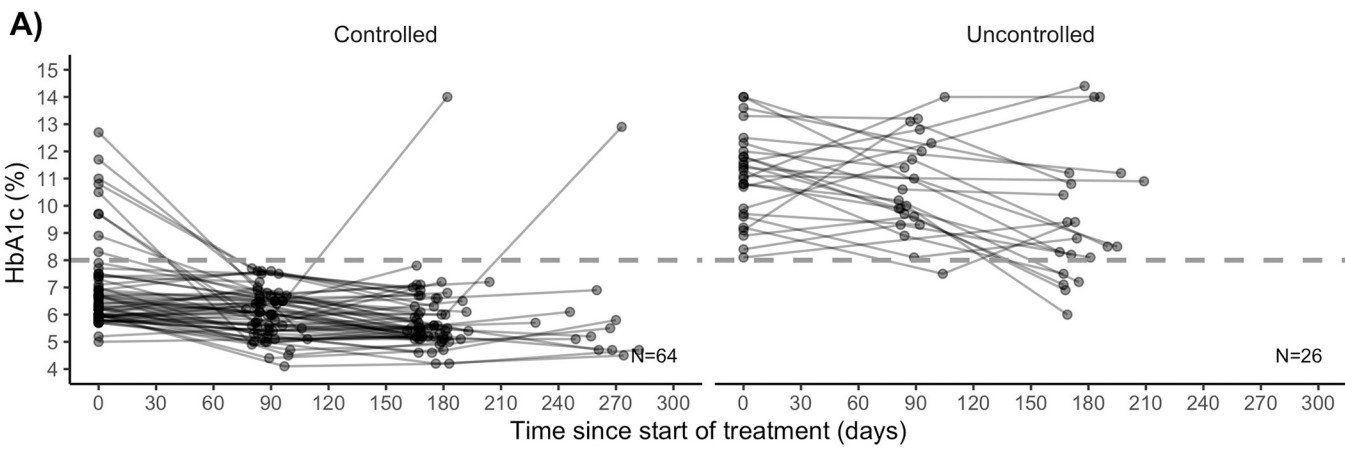

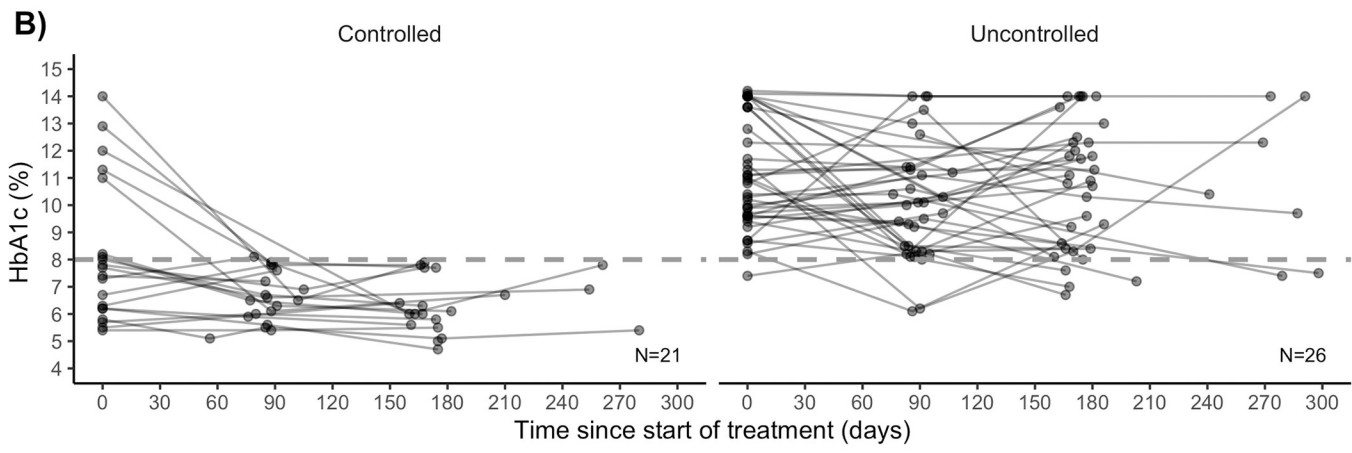

**Fig 2.** Individual glycosylated hemoglobin (HbA1c) trajectories by controlled and uncontrolled glycemia among a subset of 188 tuberculosis patients with a newly-diagnosed (A) or previously-diagnosed (B) diabetes mellitus comorbidity, Metro Manila, Cebu and Negros Occidental, Philippines, 2018–2021. HbA1c, glycosylated hemoglobin; TB, tuberculosis; DM, diabetes mellitus; tuberculosis, TB; Directly Observed Treatment for the Treatment of Tuberculosis, TB-DOTS. Footnotes: Of the 188 diabetic participants included in this study, 165 had at least two HbA1c measurements and could be assessed for having controlled or uncontrolled blood glucose. Controlled = at least two HbA1c results < 8%; uncontrolled at least two HbA1c > 8%. Fourteen of the 165 participants had an indeterminate control status by this definition and were excluded from the analysis this outcome. Individual patient lines are shown in variable shades of grey for visibility. Grey dotted line indicates 8% threshold for uncontrolled hyperglycemia.

Those with a previous versus new DM diagnosis had higher HbA1c at any time, on average (S2 Fig). The magnitude of this association was more than two times greater in those with no central obesity (coefficient 3.48, 95% CI 2.16, 4.80, P<0.001) than in those with central obesity (coefficient 0.80, 95% CI 0.26, 1.57, P = 0.043, Table 4).

The association of higher HbA1c seen in those with central obesity compared to those without in univariable analyses was only apparent among the strata of those with a new DM diagnosis in the adjusted multivariable analysis. Among new DM diagnoses, those with central obesity had higher HbA1c than those without central obesity (coefficient 1.62, 95% CI 0.72, 2.53, P = 0.009) (S2 Fig). Amongst those with a previous DM diagnosis, there was no evidence of an effect of central obesity on HbA1c (P = 0.096).

Elevated blood pressure at enrollment was associated with lower HbA1c during treatment, compared to normal blood pressure (P = 0.027, Table 4).

In the final model, participants receiving treatment for MDR-TB experienced a faster rate of decrease in HbA1c on average, over time (coefficient -0.13, 95% CI -0.20, -0.08, P<0.001)

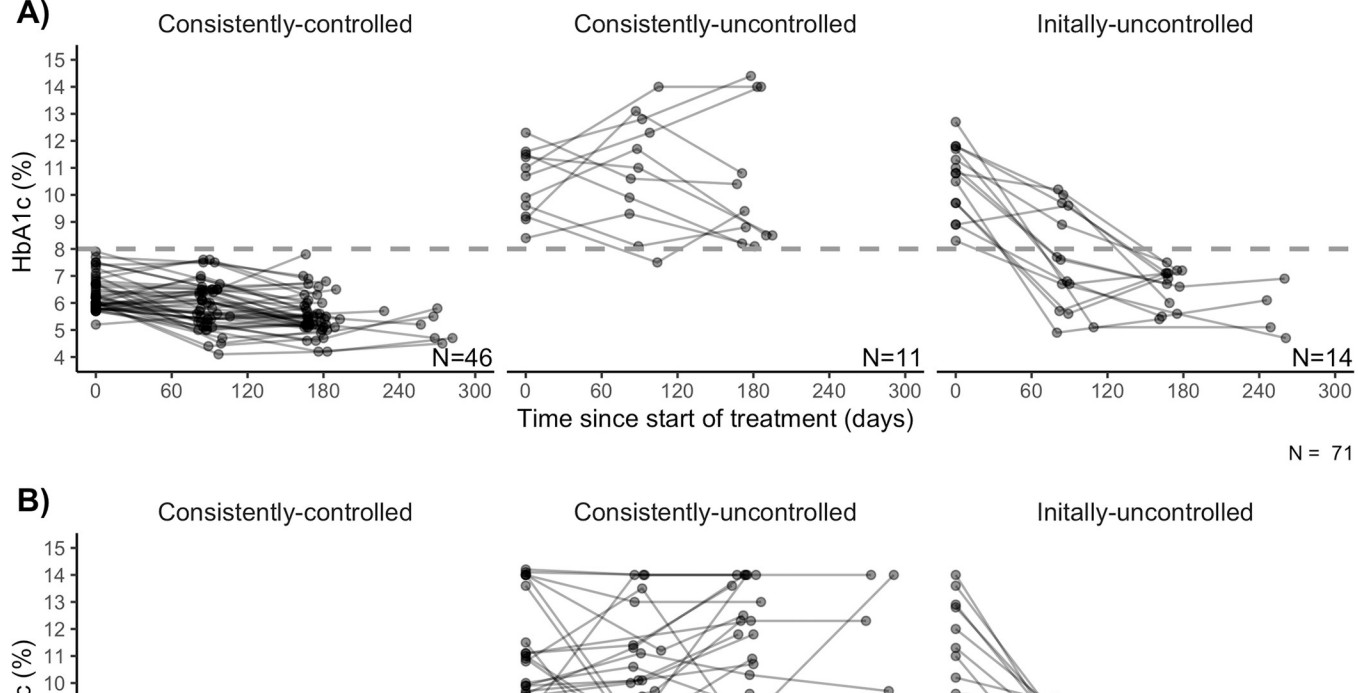

**Fig 3.** Individual glycosylated hemoglobin (HbA1c) trajectories by degree of glycemic control among a subset of 188 tuberculosis patients with a newly-diagnosed (A) or previously-diagnosed (B) diabetes mellitus comorbidity. DM, diabetes mellitus; HbA1c, glycosylated hemoglobin; TB-DOTS, Directly Observed Treatment for the Treatment of Tuberculosis. Footnotes: Of the 188 diabetic participants included in this study, 117 had at least three HbA1c measurements and could be assessed for degree of glycemic control. Controlled = all HbA1c results < 8%; initially-uncontrolled = baseline HbA1c result ≥ 8%, and all subsequent measurements < 8%; consistently-uncontrolled = all HbA1c results ≥ 8%. Four of the 117 participants had an indeterminate control status by this definition and were excluded from analyses of this outcome. Grey dotted line indicates 8% threshold for hyperglycemia.

compared to those being treated for DS-TB (coefficient -0.06, 95% CI -0.10, -0.03, P = 0.001) (S3 Fig).

## Associations with poor glycemic control

Region was significantly associated with the binary outcome of glycemic control in univariable analysis (S3 Table) and was added into the model last. The final multivariable logistic regression (Table 5) adjusted for region, central obesity, timing of DM diagnosis, use of glucose-lowering medications, and new versus relapse TB case. Previous DM diagnosis was associated with increased odds of uncontrolled glycemia (P = 0.042). Relapse TB was moderately associated with lower odds of having uncontrolled glycemia compared to a new TB case (P = 0.031). Use of glucose-lowering medications was associated with higher odds of having uncontrolled glycemia (P = 0.005). There was weak evidence of increased odds of uncontrolled glycemia among participants in Manila with a wide confidence interval (P = 0.004). Central obesity was

**Table 4. Adjusted associations with glycosylated hemoglobin (HbA1c, %) during tuberculosis (TB) treatment among 188 TB patients with a diabetes mellitus comorbidity.**

| Characteristic[a] | | Participants | HbA1c observations | Coefficient (95% CI) | P-value[b] |
|---|---|---|---|---|---|
| Effect of time[c] amongst drug-sensitive (DS) TB infection | | 147 | 369 | -0.06 (-0.10, -0.03) | 0.001 |
| Effect of time[c] among drug-resistant (MDR) TB infection | | 41 | 113 | -0.13 (-0.20, -0.08) | <0.001 |
| Blood pressure[e] | | | 482 | | 0.009 |
| Normal | | 74 | | Reference | |
| Elevated | | 16 | | -1.36 (-2.29, -0.19) | |
| Stage 1 Hypertension | | 47 | | 0.45 (-0.38, 1.23) | |
| Stage 2 Hypertension | | 35 | | -0.76 (-1.67, 0.15) | |
| Timing of DM diagnosis among those with central obesity | Newly diagnosed DM | 58 | 161 | Reference | 0.043 |
| | Previously diagnosed DM | 73 | 168 | 0.80 (0.26, 1.57) | |
| Timing of DM diagnosis among those without central obesity | Newly diagnosed DM | 41 | 118 | Reference | <0.001 |
| | Previously diagnosed DM | 16 | 35 | 3.48 (2.16, 4.80) | |
| Central obesity among those with previous DM diagnosis | No central obesity | 16 | 35 | Reference | 0.096 |
| | Central obesity | 73 | 168 | -1.06 (-2.31, 0.19) | |
| Central obesity among those with new DM diagnosis | No central obesity | 41 | 118 | Reference | 0.001 |

CI, confidence interval; DM, diabetes mellitus; HbA1c, glycosylated hemoglobin; TB, tuberculosis; TB-DOTS, TB Directly Observed Treatment Strategy.

[a] Results shown for covariates retained in the final multivariable model.

[b] Two-sided P-value from Global Wald test with small-sample adjustment for fixed effects [56].

[c] Square root of time, measured as days from start of treatment with an added constant of 0.01.

[d] Normal (Systolic blood pressure (SBP) <120 and diastolic blood pressure (DBP) <80 mm Hg); elevated (SBP 120–129 mm Hg and DBP <80 mm Hg); Stage 1 Hypertension (SBP 130–139 mm Hg and DBP 80–89 mm Hg); and Stage 2 hypertension (SBP $\geq$ 140 mm Hg and DBP $\geq$ 90 mm Hg), by the 2017 American College of Cardiology and American Heart Association guidelines [50].

[e] Based on waist-to-hip ratio >0.85 for women and >0.9 for men used by the WHO for use in diagnostic criteria for metabolic syndrome, a group of characteristics associated with increased risk of developing DM and cardiovascular disease [49].

retained in the final model *a priori*, but was not associated with glycemic control (P = 0.273). There was no evidence of between-site variation (clustering, LRT P-value = 1.000).

The final multivariable multinomial regression for degree of glycemic control (Table 5) adjusted for sex, central obesity, glucose-lowering medication use, timing of DM diagnosis, and new versus relapse TB case. With no evidence of clustering by site in the logistic regression, testing for clustering was not repeated for the multinomial regression. Relative risk of initially-uncontrolled versus controlled glycemia was higher for those with central obesity (P = 0.005) and those who used glucose-lowering medications (P = 0.001). The relative risk of consistently-uncontrolled versus controlled glycemia was also higher for those who used glucose-lowering medications (P = 0.04), for those with previously versus newly-diagnosed DM (P = 0.003), and for those with a relapse versus new TB case (P = 0.044) (Table 5). There was a non-statistically significant increased risk of initially- and consistently-uncontrolled hyperglycemia in males versus females (P>0.1).

## Discussion

This longitudinal investigation of glycemic control among diabetic patients in TB treatment identified factors associated with uncontrolled blood glucose among a cohort of Filipino patients with DM receiving treatment for TB. The results of this study showed that those who were diagnosed with DM prior to starting TB treatment were more likely to have uncontrolled blood glucose during treatment compared to those diagnosed during TB treatment. Persons with previously diagnosed DM in this study were more likely to have higher HbA1c levels on

**Table 5. Adjusted associations with poor glycemic control (controlled vs uncontrolled; initially-uncontrolled and consistently-uncontrolled vs consistently controlled) during tuberculosis treatment among a subset of 151 TB Patients with a diabetes mellitus comorbidity.**

| Characteristic | N | Logistic regression: Uncontrolled versus controlled[a] | | | N | Multinomial regression: Initially-uncontrolled versus controlled[b] | | | Multinomial regression: Consistently-uncontrolled versus controlled[2] | | |
| --- | --- | --- | --- | --- | --- | --- | --- | --- | --- | --- | --- |
| | | Uncontrolled (%) | Odds Ratio (95% CI) | Wald P-value | | Initially-uncontrolled (%) | Relative Risk Ratio[c] (95% CI) | Wald P-value | Consistently uncontrolled (%) | Relative Risk Ratio[c] (95% CI) | Wald P-value |
| Sex | | | | | | | | | | | |
| Female | 46 | 25 (54.3) | | | 32 | 4 (12.5) | Reference | 0.122 | 15 (46.9) | Reference | 0.176 |
| Male | 105 | 41 (39.0) | | | 81 | 23 (28.4) | 3.31 (0.72, 15.11) | | 18 (22.2) | 0.38 (0.09, 1.53) | |
| Region | | | | | | | | | | | |
| Cebu | 70 | 27 (38.6) | Reference | 0.013 | 55 | 13 (23.6) | | | 16 (29.1) | | |
| Negros Occidental | 66 | 26 (39.4) | 1.72 (0.72, 4.11) | | 54 | 12 (22.2) | | | 16 (29.6) | | |
| Manila | 15 | 13 (86.7) | 12.31 (2.25, 67.27) | | 4 | 2 (50.0) | | | 1 (25.0) | | |
| Central obesity[d] | | | | | | | | | | | |
| Normal | 49 | 12 (24.5) | Reference | 0.273 | 39 | 3 (7.7) | Reference | 0.005 | 8 (20.5) | Reference | 0.884 |
| Central obesity | 102 | 54 (52.9) | 1.68 (0.66, 4.30) | | 74 | 24 (32.4) | 8.76 (1.89, 40.52) | | 25 (33.8) | 1.11 (0.27, 4.45) | |
| Timing of DM diagnosis | | | | | | | | | | | |
| Newly diagnosed | 90 | 26 (28.9) | Reference | 0.042 | 71 | 14 (19.7) | Reference | 0.551 | 11 (15.5) | Reference | 0.015 |
| Previously diagnosed | 61 | 40 (65.6) | 2.50 (1.61, 6.05) | | 42 | 13 (31.0) | 1.48 (0.40, 5.52) | | 22 (52.4) | 5.14 (1.37, 19.20) | |
| Report of glucose-lowering medications during TB treatment[e] | | | | | | | | | | | |
| No use | 49 | 7 (14.3) | Reference | <0.001 | 41 | 2 (4.8) | Reference | 0.001 | 3 (7.3) | Reference | 0.003 |
| Use | 102 | 59 (57.8) | 4.78 (1.61, 14.23) | | 72 | 25 (34.7) | 10.24 (2.24, 46.81) | | 30 (41.7) | 10.24 (2.24, 46.82) | |
| New versus relapse TB case | | | | | | | | | | | |
| New | 96 | 48 (50.0) | Reference | 0.031 | 72 | 15 (20.8) | Reference | 0.708 | 27 (37.5) | Reference | 0.042 |
| Relapse | 54 | 17 (31.5) | 0.20 (0.06, 0.63) | | 41 | 12 (29.3) | 1.26 (0.37, 4.30) | | 6 (14.6) | 0.25 (0.07, 0.95) | |

CI, confidence interval; DM, diabetes mellitus; HbA1c, glycated hemoglobin; TB, tuberculosis; TB-DOTS, TB Directly Observed Treatment Strategy; GeneXpert (Cepheid), a cartridge-based nucleic acid amplification test for simultaneous rapid tuberculosis diagnosis and rapid antibiotic sensitivity test.

[a] Amongst those with ≥ 2 HbA1c results: Uncontrolled (at least two study-measured HbA1c results equal to or greater than 8%); controlled (at least two study-measured HbA1c results less than 8%). Binary glycemic control outcome is not mutually exclusive to degree of glycemic control outcome.

[b] Amongst those with ≥ 3 HbA1c results: Controlled (all HbA1c values were less than 8%); initially-uncontrolled (baseline HbA1c measurement was greater than 8%, and all subsequent measurements were less than 8%); consistently-uncontrolled (all HbA1c values were equal to or greater than 8%). Degree of glycemic control outcome is not mutually exclusive with binary glycemic control outcome.

[c] Relative risk ratio, which approximates an odds ratio when there are more than two mutually exclusive categories [57–59].

[d] Based on waist-to-hip ratio >0.85 for women and >0.9 for men used by the WHO for use in diagnostic criteria for metabolic syndrome [49].

[e] Metformin, gliclazide, or insulin use self-reported at enrollment or any point during TB treatment.

[f] Confirmed by GeneXpert (Cepheid), a cartridge-based nucleic acid amplification test for simultaneous rapid tuberculosis diagnosis and rapid antibiotic sensitivity test, or by direct sputum smear microscopy.

Pearson $\chi 2$ goodness-of-fit test for logistic regression. $\chi 2 = 23.38$, P = 0.499.

Generalized Hosmer–Lemeshow goodness-of-fit test for multinomial logistic regression. Statistic = 11.09, P = 0.527.

average at any point during treatment, and to have an overall pattern of uncontrolled blood glucose over the course of TB treatment.

One third of the persons initially suspected to have a new DM diagnosis through screening in this study may have had transient hyperglycemia. This finding supports that DM screening through TB programs should confirm diagnosis outside of the acute phase of TB treatment to avoid DM overdiagnosis [15,33].

Among those diagnosed with DM prior to TB treatment, HbA1c values at any point in treatment were higher among those without central obesity compared to those with central obesity. Among those diagnosed with DM at the start of TB treatment, HbA1c levels were higher in those with central obesity. The lack of observation of an effect of central obesity in the analysis of overall patterns of glycemic control suggest that the clinical significance of central obesity on blood glucose control may be limited, but should be investigated in larger sample sizes and more diverse populations.

Finally, patients treated for MDR-TB experienced a greater average decrease in HbA1c over time compared to those treated for DS-TB. In the analyses of patterns of blood glucose control, no associations with MDR-TB were detected; however, those with relapse TB—representing 71% (29/41) of MDR-TB cases—were significantly less likely to have a pattern of hyperglycemia during treatment. This finding may reflect that facilities for MDR-TB were implementing a different DM-specific management protocol than standard TB-DOTS programs, that patients with daily observed MDR-TB treatment were more likely to take medications, or that patients with MDR-TB were more acutely ill at the start of TB treatment and experienced a better recovery in blood glucose control with treatment. The lack of significant difference in HbA1c at the start of treatment between the groups does not lend strength to this latter explanation.

The results of these analyses suggest that a previous DM diagnosis may be associated with uncontrolled blood glucose during treatment. Diabetic persons with poorly-controlled, long-term DM are more likely to develop active TB disease and thus may be overrepresented in the sample. Sustained hyperglycemia contributes to immune dysfunction, possibly increasing risk of developing active TB disease [6]. In a recent meta-analysis including 17 studies representing countries from Asia, Latin America, Africa and Europe, the prevalence of active TB disease was more than two times greater (OR 2.05, 95% CI 1.65, 2.55, P<0.001) among diabetic patients with hyperglycemia (HbA1c>7%) compared to diabetic patients with normal glycemia [60].

Participants in this study with previously-diagnosed diabetes may have a longer duration of disease. While there are no other studies exploring this within TB-DM populations, literature from DM populations yields mixed findings about the relationship between the duration of DM and glycemic control. Some studies of patients in have identified that having DM for a longer period of time is associated with better glycemic control [61–65], while others have found a longer duration of DM disease is associated with poorer glycemic control [20,26,66].

Central obesity has been associated with insulin resistance, and is a predictor of increased mortality risk among diabetic populations [67–69]. It is also associated with poor glycemic control among diabetic populations without TB [22–26]. It is somewhat surprising that in this population with co-morbid TB, absence of central obesity was associated with higher HbA1c in those with a previous DM diagnosis more so than among those with a previous DM diagnosis who were centrally obese. Possibly, in this sub-group, loss of central fat may represent an advanced stage of disease and metabolic disruption from one or, more likely, both DM and TB. Central obesity is also a marker of metabolic syndrome, which is associated with increased risk of developing diabetes, and is a prognostic indicator of undiagnosed DM [70,71].

In persons with newly-diagnosed DM, the presence of central obesity may indicate undiagnosed, untreated DM or represent the early stages of metabolic disruption and thus associate

with increased blood glucose. Based on the results of this study, targeted support with DM management is merited for persons diagnosed with DM prior to the start of TB treatment.

National TB Programs might consider developing guidelines to non-invasively screen new TB patients for a previous DM diagnosis and other potential risk factors such a central obesity, and provide identified patients with targeted support such as routine glucose monitoring, health education and adjustment of DM medications. An intervention including these resources was tested in a randomised controlled trial among 150 pulmonary TB-DM patients at outpatient TB clinics in Indonesia, and found that TB-DM patients receiving joint TB-DM care experienced a greater decrease in HbA1c during treatment than patients randomised to receive routine DM management [44].

Some factors were not associated with blood glucose control across analyses. Both logistic regressions found relationships (P<0.05) between uncontrolled blood glucose and glucose-lowering medication use, region, sex, and new versus relapse TB case, which were not observed in the repeated HbA1c analysis. The mixed-effects linear regression did not identify a relationship between blood glucose control and glucose-lowering medication use. While glucose-lowering medications typically lower blood glucose, diabetes is a progressive condition and the use of these medications may reflect more severe disease. Only the mixed-effects regression found an association between blood pressure and blood glucose. The mixed-effects regression included more observations than the analyses of glycemic control patterns, and could identify more associations; however, the clinical implications of the repeated measures analysis are less clear. Future research is needed to understand the biological mechanism for a previous DM diagnosis contributing to high blood glucose and adverse TB treatment outcomes.

This work is presently one of the largest studies of longitudinal HbA1c data among a cohort of TB patients with comorbid DM. This contributes to the growing body of work measuring blood glucose at multiple points during TB treatment [18,30,35,44,52,72–74]. Assessing glycemic control based on data collected at the start of TB treatment may result in overdiagnosis of DM and conceal relationships between clinical outcomes and longer-term glycemic control [32,75]. In this study, longitudinal HbA1c data were used to explore temporal relationships between blood glucose control and patient characteristics during TB treatment, and to explore whether those temporal relationships were relevant to overall glycemic control during TB treatment.

There was not complete or precise enough data available on the duration of DM disease prior to enrollment to include in the analysis. Persons reporting a previous DM diagnosis self-reported the date of DM diagnosis; more than half of participants did not respond. Duration of DM disease could be an unmeasured predictor of hyperglycemia.

Glucose-lowing medication exposure over time is an important variable which was not measured, but likely an important time-dependent exposure. Measuring medication exposure over time using electronic monitoring of adherence and prescription data about medication formulation and dosage, or direct measurement of medication concentration in dried blood samples could provide useful data.

The role of HIV among TB-DM patients may be important. A high proportion of participants had an unknown HIV status in the St-ATT cohort due to refusal of testing. Though total HIV prevalence in the Philippines is low (<1%), it is rapidly increasing [76,77] and HIV coinfection in TB is 13 per 100,000 population [78].

Exclusions of participants with fewer HbA1c results could lead to bias in the analysis of associations with uncontrolled glycemia, causing over- or underestimation of the strength and association between patterns of glycemic control and exposures, or failure to detect associations during the model-building process. Furthermore, participants in Manila had fewer HbA1c results than those in other regions. To minimize the exclusion of patients with

insufficient HbA1c results, research nurses conducted targeted follow-ups for individuals with fewer than two results, making efforts to mitigate potential bias. With a higher percentage of participants in Manila having only a baseline measurement compared to the other two regions, our findings may be more reflective of participants in these two regions. Nevertheless, our recommendations relating to monitoring of diabetes control during TB treatment hold, so that it will be possible to explore how control of diabetes links to TB treatment outcomes.

The small number of participants in the outcome categories in the logistic and multinomial regressions may introduce small-sample bias away from the null. The multinomial regression in particular was intended to explore the degree of glycemic control; its findings were largely consistent with the findings in the better powered binomial regression of controlled versus uncontrolled glycemia. In the future, collecting HbA1c data from larger cohorts of TB-DM patients—from the time of enrolment and at multiple points during treatment—can offer increased power to strengthen the evidence base. We estimate that future studies would require more than 1,000 persons with TB-DM, in order to detect a doubling of the odds of TB treatment failure with poor blood glucose control, if for example 50% of individuals have poor control based on our data and 5% of those with controlled diabetes experience TB treatment failure. Such research will likely require multi-site collaboration to achieve an adequate sample size.

## Conclusion

Persons enrolling in TB treatment with previously-diagnosed DM may require increased support to manage their blood glucose during TB treatment. The presence of central obesity may indicate increased risk of hyperglycemia in people who are newly screened; prioritizing DM screening for persons with central obesity may identify people in need of greater support. This study supports a hypothesis that glucose-lowering medication use may indicate more severe disease in routine practice, and may be less effective during TB treatment without more intensive monitoring and DM treatment adjustments. National TB programs should collect data on diabetic status of TB patients and blood glucose control to with the impact of poor control of diabetes amongst TB-DM patients on TB treatment outcomes.

## Supporting information

**S1 Checklist. Inclusivity in global research.**
(DOCX)

**S2 Checklist. STROBE statement—Checklist of items that should be included in reports of cohort studies.**
(DOC)

**S1 Fig.** Distribution of individual glycosylated hemoglobin results at pre-specified measurement time points during TB treatment among 188 Patients with a with a newly-diagnosed (A) or previously-diagnosed (B) DM Comorbidity.
(DOCX)

**S2 Fig.** Predicted mean glycosylated hemoglobin (HbA1c, %) at any time, with corresponding 95% confidence interval with a newly-diagnosed (A) or previously-diagnosed (B) diabetes mellitus comorbidity, overall and stratified by central obesity by mixed-effects linear regression analysis.
(DOCX)

**S3 Fig. Predicted mean change in glycosylated hemoglobin (HbA1c, %) with corresponding 95% confidence interval by drug resistance status by mixed-effects linear regression analysis.**
(DOCX)

**S1 Table. Socio-demographic, anthropometric, TB- and DM-related characteristics at enrollment into TB treatment of the 188 study participants, comparing to the subset of 151 participants included in the binary logistic regression.** Abbreviations: BMI, Body Mass Index; DM, diabetes mellitus; HT, cartridge-based nucleic acid amplification test (Cephid), GeneXpert; hypertension; Standard deviation (SD); DSSM, Direct sputum smear microscopy (DSSM); Philippine peso (PHP), TB, Tuberculosis (TB), Hypertension (HT). a Amongst those with > 2 glycosylated hemoglobin (HbA1c) results: Uncontrolled (at least two study-measured HbA1c results equal to or greater than 8%); controlled (at least two study-measured HbA1c results less than 8%). Binary glycemic control outcome is not mutually exclusive to degree of glycemic control outcome. b Amongst those with > 3 HbA1c results: Controlled (all HbA1c values were less than 8%); initially-uncontrolled (baseline HbA1c measurement was greater than 8%, and all subsequent measurements were less than 8%); consistently-uncontrolled (all HbA1c values were equal to or greater than 8%). Degree of glycemic control outcome is not mutually exclusive with binary glycemic control outcome. c Any of the following plans: Philippines Health Insurance plan, Social Security, or Government Service Insurance. d Based on waist-to-hip ratio >0.85 for women and >0.9 for men used by the World Health Organization for use in diagnostic criteria for metabolic syndrome [49]. e Normal (Systolic blood pressure (SBP) <120 and diastolic blood pressure (DBP) <80 mm Hg); elevated (SBP 120–129 mm Hg and DBP <80 mm Hg); Stage 1 Hypertension (SBP 130–139 mm Hg and DBP 80–89 mm Hg); and Stage 2 hypertension (SBP > 140 mm Hg and DBP > 90 mm Hg), by the 2017 American College of Cardiology and American Heart Association guidelines [50]. f BMI according to WHO criteria for adults: underweight (BMI<18.5 kg/m2), normal (BMI 18.5–25.0), overweight (25.0–29.9), obese (BMI >30) [48]. g Confirmed by GeneXpert (Cepheid), a cartridge-based nucleic acid amplification test for simultaneous rapid tuberculosis diagnosis and rapid antibiotic sensitivity test, or by direct sputum smear microscopy. h Self-reported at enrollment or point during TB treatment. i After enrollment in TB treatment, report of experiencing any of the Concurrent Tuberculosis and Diabetes Mellitus Consortium (TANDEM) study DM complications [Ugarte-Gil et al. 2020]: ever lost a limb or digit not through trauma, ever had a bypass or stenting surgery in limbs, non-healing wound for three or more months, heart attack, stroke, bypass or stenting heart surgery, diagnosis of angina or heart failure, cataract or laser eye surgery, glaucoma, acquired blindness not due to trauma, difficulty seeing or disturbed vision, renal failure. Additionally, the measure captures if participant had any symptom of distal symmetrical peripheral neuropathy using the Michigan Neuropathy Screening Instrument [53].
(DOCX)

**S2 Table. Unadjusted Associations with HbA1c (%) During TB Treatment Among 188 TB Patients with a DM Comorbidity at 13 Public TB-DOTS Clinics in Metro Manila, Cebu and Negros Occidental, Philippines, 2018–2021.** Abbreviations. BMI, Body mass index; CI, confidence interval; DM, diabetes mellitus; DSSM, direct sputum smear microscopy; HbA1c, glycated hemoglobin, GSIS, Government Service Insurance System; PhilHealth, Philippines Health Insurance, SSS, Republic of the Philippines Social Security Scheme; TB, tuberculosis. Abbreviations. BMI, Body mass index; CI, confidence interval; DM, diabetes mellitus; DSSM, direct sputum smear microscopy; HbA1c, glycated hemoglobin, GSIS, Government Service Insurance System; PhilHealth, Philippines Health Insurance, SSS, Republic of the Philippines

Social Security Scheme; TB, tuberculosis. a P-value from Global Wald test with small-sample adjustment for fixed effects [56]. Bold type: P<0.1 for model building and retention criteria. b Results shown for covariates retained in the final multivariable model. c Square root of time, measured as days from start of treatment with an added constant of 0.01. d Any of the following plans: Philippines Health Insurance plan, Social Security, or Government Service Insurance. e BMI according to WHO criteria for adults: underweight (BMI<18.5 kg/m2), normal (BMI 18.5–25.0), overweight (25.0–29.9), obese (BMI >30) [48]. f Normal (Systolic blood pressure (SBP) <120 and diastolic blood pressure (DBP) <80 mm Hg); elevated (SBP 120–129 mm Hg and DBP <80 mm Hg); Stage 1 Hypertension (SBP 130–139 mm Hg and DBP 80–89 mm Hg); and Stage 2 hypertension (SBP > 140 mm Hg and DBP > 90 mm Hg), by the 2017 American College of Cardiology and American Heart Association guidelines [50]. g Based on waist-to-hip ratio >0.85 for women and >0.9 for men used by the WHO for use in diagnostic criteria for metabolic syndrome [49]. h Time-varying exposure; medication (any or individual) use reported at appointment. i After enrollment in TB treatment, report of experiencing any of the TANDEM study DM complications (Ugarte-Gil et al. 2020): ever lost a limb or digit not through trauma, ever had a bypass or stenting surgery in limbs, non-healing wound for three or more months, heart attack, stroke, bypass or stenting heart surgery, diagnosis of angina or heart failure, cataract or laser eye surgery, glaucoma, acquired blindness not due to trauma, difficulty seeing or disturbed vision, renal failure. Additionally, the measure captures if participant had any symptom of distal symmetrical peripheral neuropathy using the Michigan Neuropathy Screening Instrument [53]. j After enrollment in TB treatment, report of any of the following lifestyle changes for DM management: changes in eating habits, becoming more physically active, weight management, cutting back or quitting smoking, management stress, or decreasing alcohol intake. k Confirmed by GeneXpert (Cepheid), a cartridge-based nucleic acid amplification test for simultaneous rapid tuberculosis diagnosis and rapid antibiotic sensitivity test, or by direct sputum smear microscopy. l Adherence to TB treatment (repeated measure) determined by number of affirmative responses to an eight-question Morisky Medication Adherence Scale (MMAS-8). High adherence = 8 affirmative responses, medium adherence = 6–7, low adherence <6. No participants had a "High" MMAS score. 214 missing observations.
(DOCX)

**S3 Table. Unadjusted Associations with Uncontrolled Blood Glucose During TB Treatment among a Subset of 188 TB Patients with a DM Comorbidity at 13 Public TB-DOTS clinics in Metro Manila, Cebu and Negros Occidental, Philippines, 2018–2021.** Abbreviations: Tuberculosis (TB), Philippine peso (PHP), Hypertension (HT), Body Mass Index (BMI), Confidence interval (CI), Direct sputum smear microscopy (DSSM). a Amongst those with > 2 HbA1c results: Uncontrolled (at least two study-measured HbA1c results equal to or greater than 8%); controlled (at least two study-measured HbA1c results less than 8%). Binary glycemic control outcome is not mutually exclusive to degree of glycemic control outcome. b Amongst those with > 3 HbA1c results: Controlled (all HbA1c values were less than 8%); initially-uncontrolled (baseline HbA1c measurement was greater than 8%, and all subsequent measurements were less than 8%); consistently-uncontrolled (all HbA1c values were equal to or greater than 8%). Degree of glycemic control outcome is not mutually exclusive with binary glycemic control outcome. c Relative risk ratio, which approximates an odds ratio when there are more than two mutually exclusive categories [57,59]. d Any of the following plans: Philippines Health Insurance plan, Social Security, or Government Service Insurance. e BMI according to WHO criteria for adults: underweight (BMI<18.5 kg/m2), normal (BMI 18.5–25.0), overweight (25.0–29.9), obese (BMI >30) [48]. f Normal (Systolic blood pressure (SBP) <120

and diastolic blood pressure (DBP) <80 mm Hg); elevated (SBP 120–129 mm Hg and DBP <80 mm Hg); Stage 1 Hypertension (SBP 130–139 mm Hg and DBP 80–89 mm Hg); and Stage 2 hypertension (SBP > 140 mm Hg and DBP > 90 mm Hg), by the 2017 American College of Cardiology and American Heart Association guidelines [50]. g Based on waist-to-hip ratio >0.85 for women and >0.9 for men used by the WHO for use in diagnostic criteria for metabolic syndrome [49]. h Self-reported at enrollment or point during TB treatment. i After enrollment in TB treatment, report of experiencing any of the TANDEM study DM complications [52]: ever lost a limb or digit not through trauma, ever had a bypass or stenting surgery in limbs, non-healing wound for three or more months, heart attack, stroke, bypass or stenting heart surgery, diagnosis of angina or heart failure, cataract or laser eye surgery, glaucoma, acquired blindness not due to trauma, difficulty seeing or disturbed vision, renal failure. Additionally, the measure captures if participant had any symptom of distal symmetrical peripheral neuropathy using the Michigan Neuropathy Screening Instrument [53]. j After enrollment in TB treatment, report of any of the following lifestyle changes for DM management: changes in eating habits, becoming more physically active, weight management, cutting back or quitting smoking, management stress, or decreasing alcohol intake. k By DSSM or GeneXpert. l Adherence to TB treatment (repeated measure) determined by number of affirmative responses to an eight-question Morisky Medication Adherence Scale (MMAS-8). High adherence = 8 affirmative responses, medium adherence = 6–7, low adherence <6. No participants had a "High" MMAS score. 15/188 participants had no MMAS score recorded.
(DOCX)

## Acknowledgments

The authors would like to thank the participants and their families, staff at the health centers and TB-DOTS clinics and our team of research nurses Cristelyn Alvarez, Clarinda Berido, Michelle Caballero, Bliss Caraig, Paul Ian Flores, Romil Jeffrey Juson, Ann Lustresano, Trivon Opinion, Michelle Saavaedra and Ares Verde for their hard work and dedication to the study and the wellbeing of the participants. The authors also are grateful to Dr Naomi Saludar of San Lazaro Hospital in Manila for her support.

## Author Contributions

**Conceptualization:** Lauren Oliveira Hashiguchi, Sharon E. Cox, Tansy Edwards.

**Data curation:** Lauren Oliveira Hashiguchi, Julius Patrick Ferrer, Shuichi Suzuki, Benjamin N. Faguer, Mary Christine Castro, Sharon E. Cox, Tansy Edwards.

**Formal analysis:** Lauren Oliveira Hashiguchi, Sharon E. Cox, Tansy Edwards.

**Funding acquisition:** Lauren Oliveira Hashiguchi, Juan Antonio Solon, Mary Christine Castro, Sharon E. Cox.

**Investigation:** Lauren Oliveira Hashiguchi.

**Methodology:** Lauren Oliveira Hashiguchi, Sharon E. Cox, Tansy Edwards.

**Project administration:** Julius Patrick Ferrer, Shuichi Suzuki, Benjamin N. Faguer, Juan Antonio Solon, Mary Christine Castro, Koya Ariyoshi, Sharon E. Cox.

**Resources:** Juan Antonio Solon, Mary Christine Castro, Koya Ariyoshi, Sharon E. Cox.

**Supervision:** Sharon E. Cox, Tansy Edwards.

**Validation:** Tansy Edwards.

**Visualization:** Lauren Oliveira Hashiguchi, Sharon E. Cox, Tansy Edwards.

**Writing – original draft:** Lauren Oliveira Hashiguchi.

**Writing – review & editing:** Lauren Oliveira Hashiguchi, Julius Patrick Ferrer, Shuichi Suzuki, Benjamin N. Faguer, Juan Antonio Solon, Mary Christine Castro, Koya Ariyoshi, Sharon E. Cox, Tansy Edwards.

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
