## [Decision Letter · Decision Letter 0]

8 Jan 2024

PGPH-D-23-01570

Glycemic control during tuberculosis treatment among people with a diabetes mellitus comorbidity in the Starting Anti-Tuberculosis Treatment Cohort Study in the Philippines

Dear Dr. Edwards,

Thank you for submitting your manuscript to PLOS Global Public Health. After careful consideration, we feel that it has merit but does not fully meet PLOS Global Public Health’s publication criteria as it currently stands. Therefore, we invite you to submit a revised version of the manuscript that addresses the points raised during the review process.

We look forward to receiving your revised manuscript.

Kind regards,

Christian Wejse, MD, PhD, Professor in Global Health

Academic Editor

Journal Requirements:

1. Please include a complete copy of PLOS’ questionnaire on inclusivity in global research in your revised manuscript. Our policy for research in this area aims to improve transparency in the reporting of research performed outside of researchers’ own country or community. The policy applies to researchers who have travelled to a different country to conduct research, research with Indigenous populations or their lands, and research on cultural artefacts. The questionnaire can also be requested at the journal’s discretion for any other submissions, even if these conditions are not met.  Please find more information on the policy and a link to download a blank copy of the questionnaire here: https://journals.plos.org/gloalpublichealth/s/best-practices-in-research-reporting. Please upload a completed version of your questionnaire as Supporting Information when you resubmit your manuscript.

Additional Editor Comments (if provided):

The article does not adhere to appropriate reporting guidelines (STROBE). A completed STROBE checklist should be submitted with the revised manuscript, including a readable flow chart according to STROBE guidelines

Reviewers' comments:

Reviewer's Responses to Questions

**Comments to the Author**

1. Does this manuscript meet PLOS Global Public Health’s publication criteria? Is the manuscript technically sound, and do the data support the conclusions? The manuscript must describe methodologically and ethically rigorous research with conclusions that are appropriately drawn based on the data presented.

Reviewer #1: Yes

Reviewer #2: Yes

2. Has the statistical analysis been performed appropriately and rigorously?

Reviewer #1: Yes

Reviewer #2: Yes

3. Have the authors made all data underlying the findings in their manuscript fully available (please refer to the Data Availability Statement at the start of the manuscript PDF file)?

Reviewer #1: Yes

Reviewer #2: Yes

4. Is the manuscript presented in an intelligible fashion and written in standard English?

Reviewer #1: Yes

Reviewer #2: Yes

5. Review Comments to the Author

Reviewer #1: Manuscript review

“Glycemic control during tuberculosis treatment among people with diabetes mellitus comorbidity at the starting anti-tuberculosis treatment cohort in the Philippines”

Summary of the study

This longitudinal study examines glycemic control among Filipino adults undergoing TB treatment. The report was complete and well-written, but can be made more concise by reducing redundant sentences.

Title

I suggest removing “during tuberculosis treatment” and “comorbidity” to make the title more concise. “Glycemic control among people with diabetes mellitus in the Starting Anti-Tuberculosis Treatment Cohort Study in the Philippines”

Abstract

The abstract consists of a complete introduction, methods, results, and conclusion sections.

The results section contains many long and hard-to-read sentences.

Introduction

The introduction is one and a half pages long. I suggest restricting the introduction to a background (justification for the study conducted) and the aim of the study. Some parts of the introduction may be more suitable under the discussion section.

Methods

Table 1: Please explain why people with “HbA1c <6.5% at baseline and no subsequent HbA1c test exceeds 10%” are categorised as having transient hyperglycaemia.

Ethical approval (line 106): Please add information regarding informed consent.

I suggest removing Tables 1 and 2 and presenting the information in those tables as paragraphs to save some space and make it less like a student’s thesis with a shorter description.

Lines 142-144: Please explain why you are using WHO criteria for BMI instead of the Asia-Pacific criteria (Yoon KH, et al. Epidemic obesity and type 2 diabetes in Asia. Lancet 2006;368(9548):1681–1688).

Statistical analyses were explained very well and completely.

Results

The figures were blurry, especially Figure 1.

Table 3:

- There are mean and median age; choose one according to the data distribution.

- Some of the cells have numbers of less than 5. You may want to consider using the Fisher Exact test rather than the Chi-squared test.

- It is rather surprising that only almost 30% of the patients did not take any glucose-lowering medications during TB treatment.

I suggest including Tables S2 and S3 in the result section to understand the results of the univariable analyses before going into the model (Table 4).

Table 4:

- Please explain why glucose-lowering medication is not included in the model in Table 4.

- Are HbA1c levels at any point in treatment during treatment in Table 4 calculated as mean?

Table 5: the relative risk in the multinomial regression: consistently-uncontrolled versus controlled for those who use glucose-lowering medications is 10.24 (0.07, 0.95). This is likely a typo. Please check.

Discussion

I suggest removing the first paragraph in the discussion section (lines 412-419). The content of this paragraph belongs to the Introduction part or later part of the discussion section. The first paragraph of the discussion section should describe the main result of the paper.

Lines 472-476 “The logistic regressions found relationships (P<0.05) between uncontrolled blood glucose and glucose-lowering medication use, region, sex, and new versus relapse TB case, which were not observed in the repeated HbA1c analysis. The analysis of repeated HbA1c measures did not identify a relationship with glucose-lowering medication use.” To which part of the result does this sentence interpret?

Conclusion

Lines 521-523: “Evidence from this study suggests that use of glucose-lowering medication may not be effective during TB treatment or simply an indicator of more severe disease.” This conclusion contradicts the results shown in Table 5 and the study limitations (lines 496-499 and lines 504-509). Glucose-lowering medication may be an indication of more severe disease in routine practice and may be less effective during TB treatment unless more intensive monitoring and DM treatment adjustments are implemented (Reference number 44: Ruslami R, Koesoemadinata RC, Soetedjo NNM, Imaculata S, Gunawan Y, Permana H, et al. The effect of a structured clinical algorithm on glycemic control in patients with combined tuberculosis and diabetes in Indonesia: A randomized trial. Diabetes Res Clin Pract 2021;173:108701).

Manuscript review conclusion and recommendation

The paper has merit, and the message is important. The study design and methods were used appropriately to answer the research question. However, there are many redundancies in the results section. I suggest synchronising the statistical analyses presented in the Methods section with the Results section. Too many tables and figures were presented, resulting in a mixed message of the study. You may want to reduce the tables/figures and clarify the main message. Finally, I wish the authors the best in their research endeavours.

Reviewer #2: I would like to express my gratitude to the Editor-in-Chief for extending the invitation to review this insightful paper. Additionally, my thanks go to the authors for addressing this crucial public health issue. Please find below my comments and suggestions aimed at refining the manuscript for publication.

Rationale of the study

• While the introduction mentions the limited published data from Southeast Asia, especially the Philippines, it could provide more explicit information about the specific research gap that the current study aims to address.

• There have been several studies powered researchers conducted on this topic previously compared to your underpowered study. What distinguishes our study from the ones mentioned in the links below?

https://www.ncbi.nlm.nih.gov/pmc/articles/PMC7238456/

https://journals.plos.org/plosmedicine/article?id=10.1371/journal.pmed.1002072

https://www.sciencedirect.com/science/article/pii/S2405579423000244

Methods

• The study includes participants from clinics in different locations (urban, peri-urban, rural) and with different focuses (programmatic management of drug-resistant TB). The variability in clinic settings and TB management programs may introduce confounding variables that could impact the generalizability of the findings. The method should address how the potential site variability is controlled for in the analysis or whether it is considered as a factor.

• The process of clinic selection remains unclear. Further clarification is needed regarding the specific sampling techniques employed in this study..

Results

• The study notes that more participants in Manila had only one HbA1c measurement compared to other regions, and participants in Negros Occidental had different characteristics compared to Manila and Cebu. This variability in data collection and participant characteristics across regions may impact the generalizability of the results and should be addressed in the discussion.

Discussion

• While the discussion suggests future research directions, such as collecting data from larger cohorts, it lacks a more detailed exploration of specific research questions that need to be addressed. Providing a more nuanced discussion of potential avenues for future research would enhance the conclusion.

• The discussion provides suggestions for increased support for individuals with previously-diagnosed DM entering TB treatment and highlights the potential significance of central obesity. However, it could benefit from a more explicit discussion of how these findings could be translated into actionable strategies in clinical practice or public health interventions.

6. PLOS authors have the option to publish the peer review history of their article (what does this mean?). If published, this will include your full peer review and any attached files.

**Do you want your identity to be public for this peer review?** For information about this choice, including consent withdrawal, please see our Privacy Policy.

Reviewer #1: No

Reviewer #2: No

---

## [Editor Report · Decision Letter 1]

4 Apr 2024

Glycemic control during TB treatment among Filipinos: the Starting Anti-Tuberculosis Treatment Cohort Study

PGPH-D-23-01570R1

Dear Ms. Edwards,

We are pleased to inform you that your manuscript 'Glycemic control during TB treatment among Filipinos: the Starting Anti-Tuberculosis Treatment Cohort Study' has been provisionally accepted for publication in PLOS Global Public Health.

Best regards,

Christian Wejse, MD, PhD, Professor in Global Health

Academic Editor